# Sleep disturbance after acute coronary syndrome: A longitudinal study over 12 months

**Roland von Känel**[1]*, **Rebecca E. Meister-Langraf**[1,2], **Claudia Zuccarella-Hackl**[1], **Sarah L. F. Schiebler**[1], **Hansjörg Znoj**[3], **Aju P. Pazhenkottil**[1,4,5], **Jean-Paul Schmid**[6], **Jürgen Barth**[7], **Ulrich Schnyder**[8], **Mary Princip**[1]

**1** Department of Consultation-Liaison Psychiatry and Psychosomatic Medicine, University Hospital Zurich, University of Zurich, Zurich, Switzerland, **2** Clienia Schlössli AG, Oetwil am See, Zurich, Switzerland, **3** Department of Health Psychology and Behavioral Medicine, University of Bern, Bern, Switzerland, **4** Department of Cardiology, University Hospital Zurich, University of Zurich, Zurich, Switzerland, **5** Department of Nuclear Medicine, Cardiac Imaging, University Hospital Zurich, University of Zurich, Zurich, Switzerland, **6** Department of Internal Medicine and Cardiology, Clinic Gais AG, Gais, Switzerland, **7** Institute for Complementary and Integrative Medicine, University Hospital Zurich, University of Zurich, Zurich, Switzerland, **8** University of Zurich, Zurich, Switzerland

* roland.vonkaenel@usz.ch

**Data Availability Statement:** All relevant data are within the paper and its Supporting Information files.

## Abstract

### Background

Sleep disturbance has been associated with poor prognosis in patients with acute coronary syndrome (ACS). This study examined the course of sleep disturbance and associated factors in patients with ACS who were followed for one year.

### Methods

Study participants were 180 patients (mean age 59.6 years, 81.7% men) with ACS admitted to a tertiary hospital to undergo acute coronary intervention. Sleep disturbance was interviewer-assessed at admission (n = 180), at 3 months (n = 146), and at 12 months (n = 101) using the Jenkins Sleep Scale (JSS)-4, with a total of 414 assessments over one year. Random linear mixed regression models were used to evaluate the relationship between sociodemographic factors, cardiac diseases severity, perceived distress during ACS, comorbidities, medication, health behaviors, and sleep disturbance over time.

### Results

At admission, 3 months, and 12 months, 56.7%, 49.3%, and 49.5% of patients, respectively, scored above the mean value for sleep disturbance in the general population (JSS-4 score ≥5). There was a significant decrease in continuous JSS-4 scores over time [estimate (SE) = -0.211 (0.074), p = 0.005]. Female sex [0.526 (0.206), p = 0.012], greater fear of dying [0.074 (0.026), p = 0.004], helplessness during ACS [0.062 (0.029), p = 0.034], and a history of depression [0.422 (0.171), p = 0.015] were independently associated with higher JSS-4 scores over time.

**Funding:** This study was financially supported by grant No. 140960 from the Swiss National Science Foundation to RvK (principal investigator), JPS, US, HZ and JB. The funder had no role in study design, data collection and analysis, decision to publish, or preparation of the manuscript.

**Competing interests:** The authors have declared that no competing interests exist.

## Conclusion

Despite a decrease from admission to 3 months, sleep disturbance is prevalent in the first year after ACS. Female sex, depression history, and distress during ACS identify patients at increased risk of developing persistent sleep disturbance and may inform interventions to prevent sleep disturbance.

## Introduction

About half of patients with acute coronary syndrome (ACS) have sleep disturbance [1] assessed with validated instruments like the Pittsburgh Sleep Quality Index (PSQI) [2,3] and the Insomnia Severity Index (ISI) [4,5], capturing insomnia as a specific operationalization of sleep disturbance [6]. Sleep disturbance appears to be most severe when measured within a few days of ACS and decrease after 1–2 months, but studies with repeated assessments and longer follow-up periods are lacking [1].

Emerging evidence suggests that sleep disturbance may impact the prognosis in patients with ACS, with Mendelian randomization analyses suggesting that the link between frequent insomnia symptoms and coronary heart diseases may be causal [7]. For instance, one study in 2,246 patients with ACS showed sex-specific effects of sleep disturbance and insomnia symptoms, respectively, on short- and long-term prognosis, adjusted for a range of relevant prognostic factors [8]. In that study, poor sleep quality at least weekly and impaired awakening (i.e., difficulty rising and not feeling well rested after sleep) predicted mortality within 28 days in men. In women, disturbed sleep (including general sleep quality, difficulty falling asleep, repeated awakenings, disturbed/uneasy sleep, and premature awakenings) was associated with an increased 10-year risk of new cardiovascular events [8]. In a second study with 283 women with ACS aged 65 years and younger, a sleep quality index, including frequency of difficulties falling asleep, disturbed/restless sleep, and premature awakening, predicted recurrent coronary heart disease events over a 5-year follow-up; this association was independent of demographics, cardiovascular risk factors, and depression [9]. In a third study with 1,152 patients with ACS, sleep disturbances assessed with the Leeds Sleep Evaluation Questionnaire at baseline were associated with increased all-cause mortality after 5–12 years of follow-up even after adjustment of relevant prognostic factors [10]. The questionnaire assessed ease of getting to sleep, sleep quality, ease of morning awakening, and integrity of behavior following wakefulness [10].

Cross-sectional studies in patients with ACS have shown significant independent associations of greater sleep disturbance with younger age [5], physical comorbidity (e.g., liver disease) [4], depressive symptoms [2–4], prescribed medications for sleep [5], and greater BMI [2]. In contrast, objective measures of cardiac disease severity like the Global Registry of Acute Coronary Events risk score [2] and left ventricular ejection fraction [4] showed no significant association with sleep disturbance in previous ACS studies. However, findings are also mixed [2,5], and whether these and other variables, including perceived distress during ACS, are associated with sleep disturbance in patients with ACS longitudinally has not been examined [1]. Persistent associations of both modifiable and non-modifiable determinants of sleep disturbance over one year may reflect greater harm to the cardiovascular system than associations at a single time point in cross-sectional studies.

Gaining novel knowledge on the above questions has the potential to identify patients at risk of developing persistent sleep disturbance after ACS. Moreover, modifiable risk factors for

sleep disturbance may inform intervention studies to prevent the development of sleep distur-
bance and thereby potentially improve the prognosis of patients with ACS too. Therefore, the
aim of this observational study was to examine the course and determinants of sleep distur-
bance in patients with ACS who underwent three assessments over one year following ACS.

## Materials and methods

### Study design and participants

For this ancillary sleep disturbance study, we used data collected from patients who partici-
pated in the Myocardial Infarction-Stress Prevention Intervention (MI-SPRINT) randomized
controlled trial [11]. All participants were referred for acute coronary care intervention to the
Cardiology Department, Bern University Hospital, Switzerland. The primary aim of MI-S-
PRINT was to investigate the effects of one single session of trauma-focused counseling versus
stress counseling on the development of interviewer-rated ACS-induced posttraumatic stress
at 3 months [12]. In the present study, no difference was made between patients in the two dif-
ferent counseling groups, as the counseling intervention was not significantly associated with
sleep disturbance reported here. The MI-SPRINT trial had originally enrolled a total of 190
patients. Of these, 180 had complete data for sleep disturbance and covariates at admission
that could be analyzed for this ancillary sleep study. Of these 180 participants, 146 and 101
contributed data at 3 months and 12 months, respectively, for the analysis. The recruitment
procedure and reasons for dropouts during the 12-month study period have been detailed else-
where [13]. By far the main reason for dropouts was a lack of funding to perform the
12-month assessment [13].

Inclusion criteria for MI-SPRINT were age 18 years or older, a verified ACS, either acute
ST segment elevation MI (STEMI) or non-STEMI, and stable circulatory conditions. Further,
patients had to indicate a high level of peritraumatic distress during ACS, defined by a score of
at least 5 for pain intensity plus at least 5 for fear of dying and/or feelings of helplessness on
numeric rating scales ranging from 0–10. Exclusion criteria were emergency coronary artery
bypass grafting, severe comorbid diseases, limited orientation, cognitive impairment, current
severe depression (per the cardiologist's medical history), suicidal ideations in the prior two
weeks, insufficient knowledge of German, and participation in another trial. The ethics com-
mittee of the State of Bern, Switzerland, approved the study protocol, which was registered
under ClinicalTrials.gov (NCT01781247). All participants provided written informed consent.

### Measures

Research assistants obtained measures at admission from patient files, through a structured
medical history and a clinical interview for sleep problems. At 3 months and 12 months,
patients were invited for follow-up examinations during which research assistants assessed
current medication, health behaviors and sleep disturbance.

### Sleep disturbance

Sleep disturbance in the last month were assessed with the 4-item Jenkins Sleep Scale (JSS-4)
asking about the frequency of difficulties initiating sleep, night-time awakenings, difficulties
maintaining sleep, and non-restorative sleep [14]. Each item is rated with either "0" (not at
all), "1" (1–3 days), "2" (4–7 days), "3" (8–14 days), "4" (15–21 days), or "5" (22–31 days). The
sleep disturbance total score ranges from 0 to 20 with higher scores indicating greater sleep
disturbance. Cronbach's α was 0.70 at admission, 0.69 at 3 months, and 0.60 at 12 months,
indicating moderate internal consistency overall [15]. A recent representative survey of the

Germany population found mean values for the JSS-4 scale of 3.90, 4.66, and 5.92 for individuals aged 51–60 years, 61–70 years, and ≥71 years, respectively [16]. Because 80% of our study participants were 51 years or older, we used a JSS-4 score ≥5 as an estimate to define the prevalence of clinically relevant sleep disturbance in our sample. A sum score of 5 corresponded to the percentile rank 73 in the normative population indicating that 27% have a JSS-4 sum score of 5 or higher [16].

## Covariates

**Demographic factors.** Demographic variables included age, sex, living status (living alone yes/no), working status (full time, part time, no employment/retired), and education. Based on Switzerland's education system, education was defined as high (university graduation, including applied sciences; high school graduation/matura), medium (apprenticeship or vocational school) or low (lower than apprenticeship or vocational school) educational level [17].

**Clinical variables.** The index event (STEMI vs. non-STEMI) and the number of diseased coronary vessels, defined as luminal narrowing of ≥50%, were used as indices of ACS severity. Pain, fear of dying, and feelings of helplessness during ACS were assessed with numeric rating scales as specified under the inclusion criteria. We calculated the Charlson comorbidity index based on which we formed categories for low, intermediate, or high 10-year mortality risk [18]. To obtain information on sleep apnea, patients were asked "Have you ever been diagnosed with sleep apnea? (yes/no)". A history of depression was assessed asking patients the question "Have you ever had a depression in your life? (yes/no)". The severity of depressive symptoms was assessed at admission, at 3 and 12 months with the 13-item self-rated cognitive/affective subscale of the Beck Depression Inventory [19]. The scale yields a total score of 0–39 and comprises no sleep item. As 34 participants had missing BDI scores at admission, analyses with BDI scores were complementary only. The use of sleeping pills, including benzodiazepines (yes/no), and antidepressants (yes/no) was also recorded.

**Health behaviors.** The body mass index (BMI) was calculated based on self-disclosed weight and height ($kg/m^2$) at admission, and on measured weight and height at both follow-ups. Patients were categorized into either current smokers or former/never smokers. Physical activity per week "that makes you sweat" was categorized as none, 1-2x or 3-7x. Alcohol consumption was categorized as none, moderate or heavy (>21 standard drinks/week for men, >14 for women).

## Data analysis

Data were analyzed with SPSS 27.0 for Windows (SPSS Inc., Chicago, IL). Level of significance was p<0.05 (two-tailed). Independent t-test and Pearson chi-square test (Fisher's exact test, where appropriate) were used for group comparison for continuous and categorical variables, respectively. We conducted linear mixed effects models analysis with random intercepts and restricted maximum likelihood estimation to examine the longitudinal association of sociodemographic factors, clinical variables and health behaviors with sleep disturbance across three investigations (at hospital admission, 3 months and 12 months). Due to a non-normal distribution, JSS-4 scores (dependent variable) were square-root transformed prior to analysis. The covariates listed below were considered on an a priori basis as potential correlates or determinants of sleep disturbance. We first modelled univariable associations of each covariate with sleep disturbance over time. As a next step, we calculated multivariable models with covariates entered into models as fixed effects. These covariates were age sex, living status, education, working status, index MI, number of diseased coronary vessels, pain, fear of dying, helplessness, Charlson comorbidity index, depression history, apnea history, sleeping pills,

antidepressants, BMI, smoking status, alcohol consumption, and physical activity. Complementary univariable and multivariable models were calculated with BDI scores as an additional covariate. Medication use (sleeping pills, antidepressants) and health behaviors (BMI, smoking, alcohol consumption, and physical activity) were all entered as time varying. Random intercepts were modeled for participants. Potential effect modification of modifiable covariates on sleep disturbance was tested in exploratory analysis by entering interaction terms between these modifiable covariates and sociodemographic factors, clinical variables and health behaviors.

## Results

### Participant characteristics

Table 1 shows the baseline characteristics of the 180 study participants of whom 102 (56.7%) scored above the population norm on the JSS-4, indicating clinically relevant sleep disturbance. The mean age of the total sample was 60 years and 4 of 5 participants were male. Compared with participants without clinical sleep disturbance, those with clinically relevant sleep disturbance had perceived greater helplessness during ACS and were less likely to be current smokers.

### Change in sleep disturbance over one year and associated factors

Of the 146 and 101 participants assessed at 3 and 12 months, respectively, 72 (49.3%) and 50 (49.5%) scored ≥5 on the JSS-4. Table 2 shows the univariable and multivariable associations of participant characteristics with temporal changes in sleep disturbance expressed as continuous JSS-4 scores.

### Univariable analysis

There was a significant decrease in sleep disturbance over time, as evidenced by a significant negative estimate for "time" (p = 0.010). Moreover, in univariable analysis, female sex (p<0.001), greater fear of dying during ACS (p = 0.002), and a history of depression (p<0.001) were significantly associated with greater sleep disturbance over time, whereas living alone was associated with less sleep disturbance (p = 0.041).

### Multivariable analysis

In the fully adjusted multivariable model, the associations of "time" (p = 0.005), female sex (p = 0.012), fear of dying during ACS (p = 0.004), and depression history (p = 0.015) with sleep disturbance over time maintained significance. In addition, a higher level of helplessness during ACS was significantly associated with greater sleep disturbance over time independent of the other covariates in the model (p = 0.034).

 With original (i.e., non-transformed) JSS-4 values as the dependent variable, a participant with a 6-point higher score on the fear of dying scale and the helplessness scale had a 3.8-point higher JSS-4 total score over time, when all other covariates in the equation were held constant. Likewise, a female participant with a history of depression had a 3.9 point higher JSS-4 total score over time than a male participant without a depression history when all other covariates were held constant.

### Supplemental models with BDI scores

Next, we computed a supplemental model for the subset of n = 146 participants who had provided BDI scores at admission. As can be seen in Table 2 (last line), cognitive depressive

**Table 1. Characteristics of study participants at baseline.**

| Variables | All<br>N = 180 | JSS-4 score <5<br>N = 78 | JSS-4 score ≥5 N = 102 | P-value |
|---|---|---|---|---|
| Age, years, M (SD) | 59.6 (11.1) | 58.0 (12.5) | 60.8 (9.8) | 0.087 |
| Sex, male, n (%) | 147 (81.7) | 67 (85.9) | 80 (78.4) | 0.200 |
| Living alone, n (%) | 46 (25.6) | 19 (24.4) | 27 (26.5) | 0.748 |
| Education<br>Low, n (%)<br>Medium, n (%)<br>High, n (%) | <br>17 (9.4)<br>128 (71.1)<br>35 (19.4) | <br>5 (6.4)<br>57 (73.1)<br>16 (20.5) | <br>12 (11.8)<br>71 (69.6)<br>19 (18.6) | 0.473 |
| Working status<br>Full time, n (%)<br>Part time, n (%)<br>No employment/retired, n (%) | <br>84 (46.7)<br>24 (13.3)<br>72 (40.0) | <br>43 (55.1)<br>11 (14.1)<br>24 (30.8) | <br>41 (40.2)<br>13 (12.7)<br>48 (47.1) | 0.078 |
| STEMI, n (%) | 128 (71.1) | 52 (66.7) | 76 (74.5) | 0.250 |
| Diseased coronary vessels, M (SD) | 1.91 (0.86) | 1.94 (0.86) | 1.89 (0.87) | 0.736 |
| Pain intensity, M (SD) | 7.86 (1.65) | 7.97 (1.65) | 7.78 (1.66) | 0.435 |
| Fear of dying, M (SD) | 5.30 (2.85) | 4.94 (2.67) | 5.58 (2.97) | 0.132 |
| Helplessness, M (SD) | 5.46 (2.65) | 5.02 (2.70) | 5.80 (2.57) | 0.049 |
| Charlson comorbidity index<br>Low risk, n (%)<br>Medium risk, n (%)<br>High risk, n (%) | <br>99 (54.0)<br>45 (25.0)<br>36 (20.0) | <br>45 (57.7)<br>19 (24.4)<br>14 (17.9) | <br>54 (52.9)<br>26 (25.5)<br>22 (21.6) | 0.781 |
| Depression history, n (%) | 51 (28.2) | 18 (23.1) | 33 (32.4) | 0.171 |
| Apnea history, n (%) | 18 (10.0) | 4 (5.1) | 14 (13.7) | 0.057 |
| Sleeping pills, n (%) | 9 (5.0) | 4 (5.1) | 5 (4.9) | 1.000 |
| Antidepressants, n (%) | 16 (8.9) | 5 (6.4) | 11 (10.8) | 0.307 |
| Body mass index, kg/m$^2$, M (SD) | 27.7 (4.7) | 27.5 (5.0) | 27.8 (4.4) | 0.579 |
| Current smoker, n (%) | 80 (44.4) | 42 (53.8) | 38 (37.3) | 0.026 |
| Alcohol consumption<br>None, n (%)<br>Mild-to-moderate, n (%)<br>Heavy, n (%) | <br>32 (17.8)<br>138 (76.7)<br>10 (5.6) | <br>15 (19.2)<br>60 (76.9)<br>3 (3.8) | <br>17 (16.7)<br>78 (76.5)<br>7 (6.9) | 0.641 |
| Physical activity<br>None, n (%)<br>1-2x/week, n (%)<br>3-7x/week, n (%) | <br>84 (46.7)<br>49 (27.2)<br>47 (26.1) | <br>37 (47.4)<br>22 (28.2)<br>19 (24.4) | <br>47 (46.1)<br>27 (26.5)<br>28 (27.5) | 0.892 |
| JSS-4, score, M (SD) | 6.14 (5.21) | 1.45 (1.49) | 9.74 (4.04) | <0.001 |
| BDI, score, M (SD) | 2.73 (2.85) | 2.56 (2.69) | 2.87 (2.97) | 0.514 |

BDI, Beck Depression Inventory; JSS, Jenkins Sleep Scale; STEMI, ST-elevation myocardial infarction. BDI scores were available for a total of n = 146 study participants (n = 63 in the group with a JSS-4 score <5 and n = 83 in the group with a JSS-4 score ≥5).

symptoms showed a significant direct association with sleep disturbance over time in the uni-variable analysis (p = 0.013) but not in the multivariable model (p = 0.46). Moreover, in the multivariable model that included BDI scores as a covariate, "time" (estimate±SE = -0.237 ±0.076, p = 0.002), female sex (0.545±0.218, p = 0.014), greater fear of dying during ACS (0.066±0.028, p = 0.019), greater helplessness during ACS (0.068±0.031, p = 0.031), and depression history (0.417±0.188, p = 0.028) remained significantly associated with greater sleep disturbance over time. When depression history was removed from the multivariable model that included BDI scores, cognitive depressive symptoms did also not emerge as a sig-nificant predictor of sleep disturbance over time (p = 0.22).

**Table 2. Univariable and multivariable relations with sleep disturbance over 12 months.**

| Parameter | Univariable | | Multivariable | |
|---|---|---|---|---|
| | Estimate | SE | Estimate | SE |
| Intercept | 1.978*** | 0.075 | 2.659** | 0.988 |
| Time | -0.174* | 0.067 | -0.211** | 0.074 |
| Age | 0.003 | 0.007 | -0.006 | 0.010 |
| Female sex | 0.744*** | 0.190 | 0.526* | 0.206 |
| Living alone | -0.357* | 0.174 | 0.047 | 0.193 |
| Education | -0.255 | 0.145 | -0.182 | 0.143 |
| Working status | 0.128 | 0.081 | 0.071 | 0.113 |
| ST-elevation MI | 0.019 | 0.043 | 0.103 | 0.170 |
| Diseases coronary vessels | 0.035 | 0.088 | 0.019 | 0.087 |
| Pain | -0.008 | 0.046 | -0.058 | 0.045 |
| Fear of dying | 0.082** | 0.026 | 0.074** | 0.026 |
| Helplessness | 0.040 | 0.029 | 0.062* | 0.029 |
| Comorbidity index | 0.075 | 0.096 | 0.099 | 0.099 |
| Depression history | 0.571*** | 0.163 | 0.422* | 0.171 |
| Apnea history | 0.185 | 0.246 | 0.232 | 0.243 |
| Sleeping pills | 0.112 | 0.246 | 0.029 | 0.250 |
| Antidepressants | 0.188 | 0.208 | 0.000 | 0.212 |
| Body mass index | -0.015 | 0.015 | -0.017 | 0.016 |
| Current smoker | 0.021 | 0.142 | -0.277 | 0.158 |
| Alcohol consumption | 0.197 | 0.123 | 0.166 | 0.110 |
| Physical activity | -0.069 | 0.071 | -0.016 | 0.079 |
| BDI score | 0.057* | 0.023 | 0.018 | 0.025 |

Sleep disturbance (dependent variable) were entered as square root transformed values of Jenkins Sleep Scale-4 scores. Due to a few participants with missing information on medication and body mass index at 3 and 12 months, there were 414 counts for each variable in the multivariable model. The supplemental univariable and multivariable models with BDI scores had 385 and 375 counts, respectively.

Significance level

***P<0.001

**P<0.010

*P<0.05.

### Effect modification

Finally, we explored whether participant characteristics potentially modified the effect of both fear of dying and helplessness during ACS on sleep disturbance over time. To this end, we additionally entered interaction terms between fear of dying or helplessness during ACS and each patient characteristic separately into the multivariable model. There were no significant interactions (p-values >0.24) between fear of dying during ACS and the following variables: age, sex, living alone, education, working status, STEMI, diseased coronary vessels, Charlson comorbidity categories, sleeping pills, antidepressants, BMI, smoking, alcohol consumption, and physical activity. For helplessness during ACS, only the association with the Charlson comorbidity index was significant (p = 0.040); however, based on the multiple tests performed, this result is likely due to chance and was not investigated further.

### Discussion

The novelty of our study is a longitudinal assessment of sleep disturbance in patients with ACS at three time points over a one-year observation period. About half of our participants

reported persistent sleep disturbance of clinical relevance. This prevalence is in the range of 40–70% in previous cross-sectional ACS studies also using validated instruments [2–5]. However, comparisons of absolute values for the prevalence of sleep disturbance between studies of patient populations with ACS are complicated by the use of different assessment tools such as the PSQI, the ISI or, in our study, the JSS-4.

We found a significant decrease of continuous scores for sleep disturbance over time with the highest prevalence of clinically relevant sleep disturbance at admission, defined by a JSS-4 cut-off score ≥5. The 3- and 12-month prevalence of clinically relevant sleep disturbance was similar but lower than at admission. This finding is consistent with previous cross-sectional studies showing that average PSQI scores were higher within a few days of ACS [20] than one [2] and two months later [21]. Our study adds important insight to this sparse literature, as previous studies have not repeatedly examined sleep disturbance in the same sample of patients with ACS and beyond a 2-month period. Whether the observed decrease in sleep disturbance, which seems to set in fairly early after ACS, reflects normalized values is difficult to assess because none of these studies, including ours, had information available on sleep disturbance before ACS [1]. Large-scale prospective population-based studies with repeated assessment of sleep disturbance before and after ACS would be needed to reconcile this uncertainty. In fact, the JSS-4 covers sleep disturbance that had occurred in the month prior to ACS, but the retrospective self-assessment of sleep disturbance, including insomnia symptoms, within 48 hours of admission may become imprecise in the acute hospital setting.

We identified two modifiable and two non-modifiable factors that were associated with greater sleep disturbance over time. Specifically, female sex, a history of depression, and greater distress during ACS, defined by single-item scores for fear of dying and feelings of helplessness, were significantly associated with greater sleep disturbance in the first year after ACS. A combination of a 6-point increased score for fear of dying and helplessness on a numeric rating scale ranging from 0–10 showed the same effect size for sleep disturbance as did female sex and depression history combined. The findings suggest that perceived distress during ACS might be an underestimated factor in terms of disturbed sleep after ACS.

In exploratory analyses, we found little evidence that the effect of perceived distress during ACS on sleep disturbance was moderated by sociodemographic factors, objective markers of ACS severity, comorbidities, medications, or health behaviors. This result may have clinical implications for many patients who experience distress during ACS, regardless of sociodemographic and health characteristics. About 70% of patients experience moderate to intense fear of dying and distress during ACS [22], and higher distress during ACS has shown to be associated with an increased risk of hospitalization due to a cardiovascular event after a follow-up of three years [23]. Sleep disturbance in patients with ACS per se have also been associated with poor prognosis [6–8]. Therefore, studies of behavioral interventions to improve sleep and thus potentially prognosis in patients with ACS are warranted. Such interventions could take either a preventive approach targeting fear of dying and helplessness at the time of ACS or a therapeutic approach through cognitive behavioral therapy (CBT) for insomnia tailored to patients with ACS-related distress [24]. The European Sleep Research Society guidelines recommend CBT as a first-line treatment for chronic insomnia (i.e., duration of at least 3 months) in adults of all ages [25]. Pharmacologic interventions (i.e., benzodiazepines, benzodiazepine receptor agonists, and some sedating antidepressants) may be considered for short-term treatment of insomnia (≤4 weeks) if CBT is not sufficiently effective or not available [25]. However, it should be noted that potential side effects of these pharmacologic interventions must be considered, which may limit (e.g., mirtazapine) or even prevent (e.g., amitriptyline) their use in patients with coronary heart disease [26].

Several variables were not independently associated with sleep disturbance over time, most notably objective markers of ACS severity. This concurs with previous studies [2,4] and

supports the observation that psychological adjustment to ACS may primarily depend on the subjective perception of an ACS as threatening and traumatic [11,22,27]. The lack of an association with current depressive symptoms may be due to the low level of depressed mood, as patients who were clinically judged to have severe depression were excluded from the study. An additional explanation could be that the cognitive subscale of the BDI used in our study does not ask for sleep difficulties. This is important, as sleep disturbance is a defining symptom of minor and major depressive disorders that have a prevalence of 20–40% combined in patients after ACS [28]. However, a previous cross-sectional study showed a significant independent relationship between depressive symptoms and insomnia, even when the sleep item was removed from the BDI-II [5]. Health behaviors, although repetitively assessed, as patients are recommended to change their lifestyle after ACS [29], were also not independently associated with sleep disturbance in our study. This concurs with previous studies showing no independent associations with BMI [5], physical activity [2], and smoking [30] as well. Relatively few patients took antidepressants and sleeping pills, including benzodiazepines, and these could have been prescribed for indications unrelated to sleep, being explanations for why medications were not associated with sleep disturbance. Indeed, previous studies showed an association with prescribed medication for sleep [5], but also not with antidepressant use [4]. Low statistical power could be another explanation for why some variables did not show a significant independent association with sleep disturbance, such as medications, living alone, work status, heavy alcohol consumption, and especially a history of sleep apnea. Obstructive sleep apnea is as prevalent as 29–67% in patients with insomnia [31], but it was not objectively diagnosed in our study, so the validity of our assessment of sleep apnea must be considered questionable.

The repetitive assessment of time-variant variables, including sleep disturbance that was interviewer-assessed using a valid sleep scale, are notable strengths of our study. However, some limitations must also be mentioned. The JSS-4 is an efficient and brief preliminary screening instrument for sleep disturbance, but it is unable to cover the entire spectrum of sleep disorders, including a formal diagnosis of insomnia and daytime sleepiness [32]. The results of our study may not be generalizable to the larger population of patients with ACS. We studied a fairly well-educated, predominantly male sample with low comorbidity, originally recruited for an intervention trial on the basis of a high level of perceived distress during ACS; the latter may have influenced the prevalence, although less likely the course of sleep disturbance after ACS. Moreover, as this was a secondary analysis of data collected for a randomized controlled trial to prevent the development of ACS-induces posttraumatic stress, there was no a priori power analysis for sleep disturbances as the outcome variable. Therefore, we are unable to justify the sample size of 180 patients for this ancillary sleep study on formal statistical grounds. Depression history was not verified by a diagnosis of lifetime depression made by a health professional. We did not include a non-ACS control group, so a direct comparison of the prevalence of sleep disturbance and its determinants that are non-ACS specific was not possible. We did not perform polysomnography to assess sleep parameters objectively, which would have allowed for a comprehensive assessment of different sleep dimensions, including total sleep time, wake after sleep onset, and sleep efficiency. However, in agreement with our findings, previous studies applying polysomnography in patients with ACS found that disturbed objective sleep may improve within 6 months after ACS [1].

## Conclusions

The results of our study support the sparse literature based on cross-sectional studies showing that in patients with ACS, sleep disturbance is common and most severe in the days after ACS

with a decrease in the following three months. Female sex, history of depression, and increased fear of dying and helplessness during ACS are independently associated with greater sleep disturbance and thus may help to identify patients at risk for developing poor sleep in the first year after ACS. As potentially modifiable factors and easy-to-asses distress measures at admission, fear of dying and feelings of helplessness during ACS may inform intervention studies to prevent patients from developing sleep disturbance and thereby potentially adverse clinical outcomes.

## Supporting information

**S1 Dataset.**
(DOCX)

**S2 Dataset.**
(DOCX)

**S3 Dataset.**
(DOCX)

## Author Contributions

**Conceptualization:** Roland von Känel, Hansjörg Znoj, Jean-Paul Schmid, Mary Princip.

**Data curation:** Rebecca E. Meister-Langraf, Mary Princip.

**Formal analysis:** Roland von Känel.

**Funding acquisition:** Roland von Känel, Hansjörg Znoj, Jean-Paul Schmid, Jürgen Barth, Ulrich Schnyder.

**Investigation:** Roland von Känel, Rebecca E. Meister-Langraf, Mary Princip.

**Methodology:** Roland von Känel, Claudia Zuccarella-Hackl, Sarah L. F. Schiebler, Hansjörg Znoj, Aju P. Pazhenkottil, Jean-Paul Schmid, Jürgen Barth, Mary Princip.

**Project administration:** Rebecca E. Meister-Langraf.

**Supervision:** Hansjörg Znoj, Aju P. Pazhenkottil, Jean-Paul Schmid, Jürgen Barth, Ulrich Schnyder.

**Validation:** Claudia Zuccarella-Hackl, Sarah L. F. Schiebler, Aju P. Pazhenkottil, Jean-Paul Schmid, Jürgen Barth, Ulrich Schnyder.

**Writing – original draft:** Roland von Känel.

**Writing – review & editing:** Rebecca E. Meister-Langraf, Claudia Zuccarella-Hackl, Sarah L. F. Schiebler, Hansjörg Znoj, Aju P. Pazhenkottil, Jean-Paul Schmid, Jürgen Barth, Ulrich Schnyder, Mary Princip.

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
