## [Decision Letter · Decision Letter 0]

22 Apr 2022

PONE-D-22-04546Sleep disturbance after acute coronary syndrome: A longitudinal study over 12 monthsPLOS ONE

Dear Dr. von Känel,

Thank you for submitting your manuscript to PLOS ONE. After careful consideration, we feel that it has merit but does not fully meet PLOS ONE’s publication criteria as it currently stands. Therefore, we invite you to submit a revised version of the manuscript that addresses the points raised during the review process.

We look forward to receiving your revised manuscript.

Kind regards,

Carmine Pizzi

Academic Editor

PLOS ONE

Journal Requirements:

“This study was financially supported by grant No. 140960 from the Swiss National Science Foundation. Additional support came from the Teaching and Research Directorate, Bern University Hospital, Switzerland.”

“This study was financially supported by grant No. 140960 from the Swiss National Science Foundation to RvK (principal investigator), JPS, US, HZ and JB. The funder had no role in study design, data collection and analysis, decision to publish, or preparation of the manuscript.”

Reviewers' comments:

Reviewer's Responses to Questions

**Comments to the Author**

1. Is the manuscript technically sound, and do the data support the conclusions?

Reviewer #1: Partly

Reviewer #2: Yes

2. Has the statistical analysis been performed appropriately and rigorously? 

Reviewer #1: Yes

Reviewer #2: Yes

3. Have the authors made all data underlying the findings in their manuscript fully available?

Reviewer #1: Yes

Reviewer #2: Yes

4. Is the manuscript presented in an intelligible fashion and written in standard English?

Reviewer #1: Yes

Reviewer #2: Yes

5. Review Comments to the Author

Reviewer #1: Based on cross-sectional studies and this ancillary data analysis, it appears that patients with ACS, sleep disturbance is common and most severe in the days after ACS with a decrease in the following three months. Female sex, history of depression, and increased fear of dying and helplessness during ACS are independently associated with greater sleep disturbance. Potentially modifiable factors were examined as well. Supplemental models and effect modifier analyses were done reasonably well. Table 2 appears to summarize the results.

As a minor note, the presentation lacks design considerations given the 190 were enrolled with 180 having complete data. There should be some statistical justification for the sufficiency of 180 subjects in this analysis.

Reviewer #2: The purpose of this study was to describe the course of sleep disturbance and associated factors in patients with ACS who were followed for one year.

The manuscript is interesting. This topic subject is original and the long follow-up add important knowledge on this field

Sleep disorders have emerged as a major cardiovascular risk factor in recent years and the pandemic has revealed how these disorders have increased as a response to stress. The manuscript offers a very interesting follow-up

Methods are well described.

Results are good and I like tables

Sleep disturbances are a modifiable risk factor through non-drug therapy (increased physical activity, relaxation techniques, adequate nutrition) or drug therapy. A comment on this in the discussion would improve the manuscript

6. PLOS authors have the option to publish the peer review history of their article (what does this mean?). If published, this will include your full peer review and any attached files.

Reviewer #1: No

Reviewer #2: **Yes: **Anna Vittoria Mattioli

---

## [Author Response · Author response to Decision Letter 0]

22 Apr 2022

Response to Reviewers 

PONE-D-22-04546: Sleep disturbance after acute coronary syndrome: A longitudinal study over 12 months

Review Comments to the Author

Reviewer #1: Based on cross-sectional studies and this ancillary data analysis, it appears that patients with ACS, sleep disturbance is common and most severe in the days after ACS with a decrease in the following three months. Female sex, history of depression, and increased fear of dying and helplessness during ACS are independently associated with greater sleep disturbance. Potentially modifiable factors were examined as well. Supplemental models and effect modifier analyses were done reasonably well. Table 2 appears to summarize the results.

We thank the Reviewer for the favorable appraisal of our manuscript.

Comment 1

As a minor note, the presentation lacks design considerations given the 190 were enrolled with 180 having complete data.

Response

We now clarified that the MI-SPRINT trial had originally enrolled a total of 190 patients and that 180 had complete data that could be analyzed for this sleep study. P 5, lines 106-109:

The MI-SPRINT trial had originally enrolled a total of 190 patients. Of these, 180 had complete data for sleep disturbance and covariates at admission that could be analyzed for this ancillary sleep study. 

Comment 2

There should be some statistical justification for the sufficiency of 180 subjects in this analysis.

Response

We believe that the reviewer is asking for a power analysis to justify the sample size of our study. However, because this was a secondary analysis of data collected for a randomized controlled trial with clinically relevant ACS-induced posttraumatic stress as the primary outcome, we had not preregistered a power analysis with sleep disturbance as the outcome. Moreover, a post-hoc power analyses as an attempt to determine whether the sample size of a secondary data analysis is adequate for a proposed analysis are commonly discouraged as they may produce misleading results and are logically invalid (see, for instance, Dziak et al, 2020):

Dziak JJ, Dierker LC, Abar B. The Interpretation of Statistical Power after the Data have been Gathered. Curr Psychol. 2020 Jun;39(3):870-877. doi: 10.1007/s12144-018-0018-1.

However, we now mention as a limitation of our study that we had not performed a power analysis that might have justified the sample size of our ancillary sleep study. P 17, lines 362-366:

Moreover, as this was a secondary analysis of data collected for a randomized controlled trial to prevent the development of ACS-induces posttraumatic stress, there was no a priori power analysis for sleep disturbances as the outcome variable. Therefore, we are unable to justify the sample size of 180 patients for this ancillary sleep study on formal statistical grounds. 

Reviewer #2: The purpose of this study was to describe the course of sleep disturbance and associated factors in patients with ACS who were followed for one year. The manuscript is interesting. This topic subject is original and the long follow-up add important knowledge on this field. Sleep disorders have emerged as a major cardiovascular risk factor in recent years and the pandemic has revealed how these disorders have increased as a response to stress. The manuscript offers a very interesting follow-up. Methods are well described. Results are good and I like tables.

We thank the Reviewer for the favorable appraisal of our manuscript.

Comment 1

Sleep disturbances are a modifiable risk factor through non-drug therapy (increased physical activity, relaxation techniques, adequate nutrition) or drug therapy. A comment on this in the discussion would improve the manuscript.

Response

We thank the Reviewer for this valuable point. We added the following information to the discussion, P 16, lines 320-327:

The European Sleep Research Society guidelines recommend CBT as a first-line treatment for chronic insomnia (i.e., duration of at least 3 months) in adults of all ages [25]. Pharmacologic interventions (i.e., benzodiazepines, benzodiazepine receptor agonists, and some sedating antidepressants) may be considered for short-term treatment of insomnia (≤4 weeks) if CBT is not sufficiently effective or not available [25]. However, it should be noted that potential side effects of these pharmacologic interventions must be considered, which may limit (e.g., mirtazapine) or even prevent (e.g., amitriptyline) their use in patients with coronary heart disease [26].

---

## [Decision Letter · Decision Letter 1]

24 May 2022

Sleep disturbance after acute coronary syndrome: A longitudinal study over 12 months

PONE-D-22-04546R1

Dear Dr. von Känel,

We’re pleased to inform you that your manuscript has been judged scientifically suitable for publication and will be formally accepted for publication once it meets all outstanding technical requirements.

Kind regards,

Carmine Pizzi

Academic Editor

PLOS ONE

---

## [Editor Report · Acceptance letter]

26 May 2022

PONE-D-22-04546R1 

Sleep disturbance after acute coronary syndrome: A longitudinal study over 12 months 

Dear Dr. von Känel:

I'm pleased to inform you that your manuscript has been deemed suitable for publication in PLOS ONE. Congratulations! Your manuscript is now with our production department. 

Kind regards, 

on behalf of

Prof Carmine Pizzi 

Academic Editor

PLOS ONE